# A 4-Week Diet Low or High in Advanced Glycation Endproducts Has Limited Impact on Gut Microbial Composition in Abdominally Obese Individuals: The deAGEing Trial

**DOI:** 10.3390/ijms23105328

**Published:** 2022-05-10

**Authors:** Armand M. A. Linkens, Niels van Best, Petra M. Niessen, Nicole E. G. Wijckmans, Erica E. C. de Goei, Jean L. J. M. Scheijen, Martien C. J. M. van Dongen, Christel C. J. A. W. van Gool, Willem M. de Vos, Alfons J. H. M. Houben, Coen D. A. Stehouwer, Simone J. M. P. Eussen, John Penders, Casper G. Schalkwijk

**Affiliations:** 1Department of Internal Medicine, Maastricht University Medical Center, 6229 ER Maastricht, The Netherlands; armand.linkens@maastrichtuniversity.nl (A.M.A.L.); petra.niessen@maastrichtuniversity.nl (P.M.N.); j.scheijen@maastrichtuniversity.nl (J.L.J.M.S.); b.houben@maastrichtuniversity.nl (A.J.H.M.H.); cda.stehouwer@mumc.nl (C.D.A.S.); 2CARIM School for Cardiovascular Diseases, Maastricht University, 6229 ER Maastricht, The Netherlands; simone.eussen@maastrichtuniversity.nl; 3Department of Medical Microbiology, Maastricht University Medical Center, 6229 ER Maastricht, The Netherlands; n.vanbest@maastrichtuniversity.nl (N.v.B.); j.penders@maastrichtuniversity.nl (J.P.); 4NUTRIM School of Nutrition and Translational Research in Metabolism, Maastricht University, 6229 ER Maastricht, The Netherlands; 5Department of Epidemiology, Maastricht University, 6229 HA Maastricht, The Netherlands; nicole.wijckmans@maastrichtuniversity.nl (N.E.G.W.); erica.degoei@maastrichtuniversity.nl (E.E.C.d.G.); mcjm.vandongen@maastrichtuniversity.nl (M.C.J.M.v.D.); c.vangool@maastrichtuniversity.nl (C.C.J.A.W.v.G.); 6Laboratory of Microbiology, Wageningen University, 6708 WE Wageningen, The Netherlands; willem.devos@wur.nl; 7Human Microbiome Research Program, Faculty of Medicine, University of Helsinki, FI-00014 Helsinki, Finland; 8CAPHRI School for Care and Public Health Research Unstitute, Maastricht University, 6229 ER Maastricht, The Netherlands

**Keywords:** dietary advanced glycation end products, dietary dicarbonyls, UPLC-MS/MS, RCT, gut microbiota, 16S rRNA, alpha diversity, beta diversity, differential abundance

## Abstract

Dietary advanced glycation endproducts (AGEs), abundantly present in Westernized diets, are linked to negative health outcomes, but their impact on the gut microbiota has not yet been well investigated in humans. We investigated the effects of a 4-week isocaloric and macronutrient-matched diet low or high in AGEs on the gut microbial composition of 70 abdominally obese individuals in a double-blind parallel-design randomized controlled trial (NCT03866343). Additionally, we investigated the cross-sectional associations between the habitual intake of dietary dicarbonyls, reactive precursors to AGEs, and the gut microbial composition, as assessed by 16S rRNA amplicon-based sequencing. Despite a marked percentage difference in AGE intake, we observed no differences in microbial richness and the general community structure. Only the *Anaerostipes* spp. had a relative abundance >0.5% and showed differential abundance (0.5 versus 1.11%; *p* = 0.028, after low- or high-AGE diet, respectively). While the habitual intake of dicarbonyls was not associated with microbial richness or a general community structure, the intake of 3-deoxyglucosone was especially associated with an abundance of several genera. Thus, a 4-week diet low or high in AGEs has a limited impact on the gut microbial composition of abdominally obese humans, paralleling its previously observed limited biological consequences. The effects of dietary dicarbonyls on the gut microbiota composition deserve further investigation.

## 1. Introduction

Dietary advanced glycation endproducts (AGEs), a heterogeneous group of sugar-modified proteins, are abundantly present in heated foods [1]. A diet high in AGEs has been linked to negative health outcomes, such as weight gain [2], increased risk of several cancer types [3,4,5,6], and insulin resistance, although results for the latter are inconsistent [7,8]. However, it is currently not understood how dietary AGEs elicit biological effects. Potentially, this could be mediated by modulation of the gut microbiota composition. Foods and dietary patterns are well known to be able to influence gut microbiota [9], which in turn are increasingly recognized as key players in the development of obesity [10] and cancer [11]. Although the metabolism of (individual) dietary AGEs is still largely unknown, early estimations using ELISA techniques suggest that only 10% of dietary AGEs are absorbed into circulation, so that the majority should pass through the colon [12]. Here, dietary AGEs have the potential to act as substrates for gut microbes, ultimately altering their composition. Indeed, several in vitro studies report that AGEs may be selectively metabolized by certain gut microbes [13,14]. In line with this, a baked chow diet high in AGEs has been shown to alter gut microbial composition in mice [15,16,17] and to reduce caecal short chain fatty acid concentrations [15,18]. However, results concerning specific genera are not consistent. Extrapolating these findings to humans is difficult, not only because of inter-species differences in gut microbiota composition but also because baking may decrease the nutritive value of proteins and micronutrients, but may also improve antioxidant capabilities of food [19].

So far, only two randomized controlled trials (RCTs) have addressed the effects of a low- or high-AGE diet on gut microbiota content in humans, and both were limited by their small sample size and highly selective patient groups [20,21]. In 20 male adolescents with a mean age of 12 years, comparisons of a two-week diet high and low in AGEs led to a reduction in lactobacilli and an expansion in enterobacteria [20]. In 20 peritoneal dialysis patients, a one-month diet low in AGEs compared to a habitual diet high in AGEs led to a reduction in *Prevotella copri* and *Bifidobacterium animalis*, while there was an expansion of *Alistipes indistinctus*, *Clostridium hatewayi, Clostridium citroniae*, and *Ruminococcus gauvreauii* [21]. However, low- and high-AGE diets in these trials were achieved by modulating food preparation methods through heat, which may lead to similar limitations as described above. Furthermore, these findings cannot be extrapolated to the general population. Of note, it was observed in the microbiota development that increased intake of fructoselysine via formula milk in early life is associated with increased levels of *Intestinimonas*-like bacteria that are known to convert this Amadori product into butyrate [22,23].

The consequences of dietary dicarbonyls on gut microbiota also deserve further investigation. These small and highly reactive molecules, also present in foods [24], may lead to rapid formation of AGEs within the body. Despite their high reactivity, simulated gastrointestinal digestion experiments suggest that dietary dicarbonyls may pass through the stomach and reach the colon largely unaltered [25], where they may also exert effects on the gut microbiota. Manuka honey, highest in the dicarbonyl methylglyoxal of all measured food items [26], shows strong antibacterial properties [27,28]. However, relationships between gut microbial composition and dicarbonyls from the habitual diet have not yet been investigated, as an extensive dietary dicarbonyls database has only recently been developed [26].

As such, we investigated the effects of a specifically designed 4-week diet low or high in AGEs on the gut microbiota composition of abdominally obese but otherwise healthy individuals in a parallel-design double blind RCT. Secondly, we also investigated cross-sectional associations between the habitual intake of dicarbonyls and the gut microbiota composition in these individuals prior to the dietary intervention.

## 2. Results

### 2.1. Baseline Characteristics

Of 82 enrolled participants, 34 participants allocated to the low-AGE diet and 36 participants allocated to the high-AGE diet collected stool samples during both their baseline and follow-up visit. Reasons for missing data were mainly the inability to collect stool within the 24-h timeframe before their lab visit (*n* = 4) or a dropout from the intervention (*n* = 10) (Figure 1). Dropouts occurred due to reasons unrelated to the dietary intervention. By design, participants were abdominally obese but otherwise healthy (Table 1). The Firmicutes/Bacteriodetes ratio of all participants is shown in Appendix A.

### 2.2. Dietary Intake during the Intervention

As published previously [8], the intake of dietary AGEs during the 4-week intervention period was increased 2.5–5.2 fold in the high-AGE group in comparison to the low-AGE group, and this difference was confirmed by significantly higher levels of free AGEs in the plasma and urine after the high-AGE diet compared to the low-AGE diet. (Table 2). By comparison, the habitual intake of AGEs in this cohort, assessed by a FFQ, was 4.07 ± 1.71 mg/day for CML, 3.83 ± 1.78 mg/day for CEL, and 26.98 ± 10.22 mg/day for MG-H1. Importantly, daily energy intake during the intervention was not statistically different between groups, and there was no difference in body weight after the intervention (mean difference (kg) [95% CI] for a low- vs. high-AGE diet of −0.4 [−1.09,0.3], *p =* 0.31). Although the intervention diets were by design matched for macronutrients, the actual intake of energy as fat and carbohydrates was marginally but statistically different between groups (Table 2). One participant, allocated to the low-AGE diet was deemed noncompliant based on a large increase in free AGEs in their plasma (37% for CML, 197% for CEL, and 568% for MG-H1) [8]. However, as per the intention-to-treat design, this participant was included in all analyses. Despite this, we performed sensitivity analyses for all intervention-related outcomes excluding this participant. In the case of different results, this was noted in the respective sections.

### 2.3. Microbial Richness and Diversity Following the Low- and High-AGE Diet

First, we determined the effect of the low- versus the high-AGE diet on the observed microbial richness and diversity, expressed as the Shannon index. Although a trend of decreased microbial richness after the low-AGE diet was observed, no differences after the low- versus the high-AGE diet were found (Figure 2). Likewise, there was no difference in microbial diversity (Shannon index) after the low- versus the high-AGE diet (Figure 2).

### 2.4. Microbial Community Structure Following the Low- and High-AGE Diet

Next, we investigated changes in the general community structure expressed as either the Bray–Curtis dissimilarity or the Aitchison distance. Based on PCoA plots of Bray–Curtis dissimilarity, we observed no apparent difference in the overall microbial composition after the low- versus the high-AGE diet (Figure 3, upper row). PERMANOVA analysis revealed a borderline significant difference in centroids between groups (*p =* 0.078), which was less apparent at the baseline (*p =* 0.341). To identify other potential diet-induced changes in the overall microbial composition, we next tested whether the intra-individual change in the Bray–Curtis dissimilarity and the overall Bray–Curtis dissimilarity of all participants within a group were different after the low- versus the high-AGE diet. However, there was no difference in the intra-individual change in the Bray–Curtis dissimilarity due to the low- or high-AGE diet (Appendix A). Although the within-group beta diversity was statistically significantly lower after the low- versus the high-AGE diet, indicating a more similar microbial composition, the overall difference between groups was only −0.01 [−0.02,−0.00] (Appendix A).

We also visualized the general community structure with PCA plots of the Aitchison dissimilarity, showing taxa underlying the ordination. This revealed that the taxa driving a potential difference in the general community structure after both diets (including *Oscillispiraceae Family, Prevotella*, and *Holdemanella*) already tended to do so before allocation of the dietary intervention. (Figure 3, lower row).

### 2.5. Differentially Abundant Genera after the Low- and High-AGE Diet

Next, we determined specific changes in the gut microbial composition by comparing the relative abundance of all genera after the low- versus the high-AGE diet using beta-binomial regression. After adjusting for sex and age, and disregarding differentially abundant genera at the baseline, the low- versus the high-AGE diet led to an enrichment in the genera *Tyzzerella* and *Family_XII_UCG-001,* and to a contraction in the genera *Negativibaccillus*, *Oscillibacter*, and *Anaerostipes* (Appendix A). Of these differentially abundant genera, *Anaerostipes* had the highest median relative abundance (0.55% [0.31,1.29] vs. 1.11% [0.67,2.05] after the low- versus the high-AGE diet) (Figure 4). Furthermore, these comparisons lost statistical significance after correction for multiple testing (all *p*-values > B-H critical *q* of 0.004). Of note, after exclusion of the non-compliant participant, there was an enrichment in the genus *Christensenellaceae_R-7 Group* after the low- versus the high-AGE diet (median relative abundance (%) [95% CI] of 0.70% [0.40,2.25] vs. 0.62% [0.28,1.27]), while all other comparisons were materially unchanged (Figure 4).

### 2.6. Associations between Habitual Intake of Dicarbonyls and Gut Microbial Composition

Finally, we also investigated associations between the habitual intake of dicarbonyls, assessed before the allocation to the low- and high-AGE interventions, and alpha diversity, beta diversity, and microbial abundance. The mean habitual intake of dicarbonyls was 3.74 ± 1.49 mg/day for MGO and 3.59 ± 1.42 mg/day for GO, and the median for 3-DG was 14.54 [10.37,24.81] mg/day. There was a trend of decreased microbial diversity (Shannon index) with a higher intake of 3-DG (standardized beta (95% CI) of −0.07 [−0.14,0.01] after adjusting for age and sex. Although additionally adjusting for energy intake and the Dutch Healthy Diet Index further widened the confidence interval, the effect size remained similar (−0.07 [−0.18,0.03]) (Appendix A). Greater intake of 3-DG was not associated with microbial richness. In line with this, the habitual intake of dietary MGO and GO were also not associated with microbial richness or diversity (Shannon index) (Appendix A). Likewise, PERMANOVA analysis revealed that the habitual intake of these dicarbonyls did not predict variation in the Bray–Curtis dissimilarity (*p =* 0.54 for MGO, *p =* 0.64 for GO, and *p =* 0.60 for 3-DG, data not shown).

In contrast, beta-binomial regression revealed several associations between the habitual intake of 3-DG and genera abundance. After adjusting for age, sex, energy intake, and the Dutch Healthy Diet index, and with correction for multiple testing, a higher habitual intake of 3-DG was positively associated with *Anaerostipes*, *Fusicatenibacter*, and *Tyzzerella*, and inversely associated with the *Adlercreutzia*, *Family_XIII_AD3011 group* and the *Eubacterium Siraeum group* (*p*-values < B-H critical *q* of 0.005) (Appendix A). A higher habitual intake of GO was inversely associated with *Colidextribacter*, *Gastranaerophilales order*, the *Ruminoccocus gnavus group*, and *Veilonella* (*p*-values < B-H critical *q* of 0.003) (Appendix A). After correction for multiple testing, the habitual intake of MGO was not associated with genera abundance (all *p*-values > B-H critical *q* of 0.005) (Appendix A). Relative abundances of these genera are shown in Appendix A. Interestingly, the observed associations were not shared between dietary dicarbonyls.

## 3. Discussion

In the present double-blind parallel-design RCT, we show limited effects of a specifically designed 4-week diet low or high in AGEs on the gut microbial composition of 70 abdominally obese individuals, despite a large difference in AGE intake between the groups. In contrast, the habitual intake of dicarbonyls, reactive precursors of AGEs, was associated with both lower and higher abundances of several genera.

The intake of AGEs in our low- and high-AGE diet, based on regular food items and not solely on different preparation methods, was markedly different (157% for CML, 420% for CEL, and 257% for MG-H1). Despite this, energy intake was similar between groups, and there were no large differences in macronutrient intake. Compliance with the intervention was further confirmed by the increase in AGEs in plasma and urine after the high- compared to the low-AGE diet [8]. Several meta-analyses suggest that a diet high in AGEs is linked to negative biological effects [29,30,31]. The biological mechanisms underlying dietary AGE-induced effects are largely unknown since mechanisms occurring with the endogenous formation of AGEs cannot be directly extrapolated. From the observations that most dietary AGEs reach the colon undigested [12] and that diet is a key modulator of the gut microbial composition [9], an interplay between dietary AGEs and gut microbes has been proposed as a contributing mechanism to the harmful effects of dietary AGEs. Indeed, modulation of the gut microbial composition of mice after a baked chow diet high in AGEs has been shown [15,16,17,20].

Despite these findings in mice, we observed no differences in microbial diversity and richness (alpha diversity) or overall gut microbial composition (beta diversity) after the low- versus the high-AGE diet. In agreement with our findings, Yacoub et al. showed no difference in the Shannon–Wiener index after a one-month diet low or habitual with regard to dietary AGEs in 20 peritoneal dialysis patients [21]. In contrast, they showed a separation in the gut microbiota composition after both diets using dimension reduction analysis, although this is not directly comparable to our measures of beta-diversity. In line with our unchanged alpha and beta diversities, our low-AGE diet compared to the high-AGE diet only led to changes in the abundance of five genera with a low relative abundance. None of these genera were changed in the two other human trials [20,21].

In the main analyses of the deAGEing trial, we observed no effects of diets low or high in AGEs on insulin sensitivity, clearance, and secretion, vascular function, overall inflammation, or biochemical parameters [8]. Our findings, therefore, not only suggest limited consequences of a 4-week diet low or high in AGEs on the gut microbial composition, but also of limited biological effects overall. As such, our results are in apparent disagreement with the two aforementioned trials in humans [20,21], but also with trials in mice [15,16,17,32,33]. Differences in the study design could explain the discrepancies between our data and both the human and mice trials. Most importantly, the external validity of these findings is limited, as there are large inter-species differences in the gut microbiota composition between mice and humans, but also within the study populations of the human trials (i.e., abdominally obese but healthy adults vs. adolescent boys [20] and peritoneal dialysis patients [21]). Unfortunately, a direct comparison in AGE intake between these studies is not possible, as AGEs in food were not determined by the gold standard of mass spectrometry in these other studies. In addition, it is unclear whether the results of previous studies can solely be ascribed to dietary AGEs. The modulation of dietary AGEs in these studies was induced by differences in cooking methods, which may also lead to the formation of other Maillard reaction products such as acrylamide [34], a carcinogenic compound in mice [35], or lead to the degradation of heat-sensitive vitamins. To avoid such limitations, we have used a specifically designed dietary intervention based on regular food items in our gold-standard dietary AGE database. Furthermore, we used sophisticated statistical methods for our differential abundance analysis. “Standard” non-parametric statistical tests, such as the Mann–Whitney U or the Wilcoxon rank-sum test, do not appropriately take into account the compositional nature of the data and are subject to inflated false discovery rates [36,37,38]. However, a limitation of our trial, compared to the animal studies, is that a 4-week diet may be too short to result in changes in the gut microbial composition. Other general limitations include the analysis of 16S rRNA instead of metagenomics sequencing and the focus on markers based on relative abundance instead of absolute abundance.

The small changes in genera abundance after the low- versus the high-AGE diet in the present study are unlikely to have large health implications. Additionally, all comparisons became statistically non-significant after adjusting for multiple testing. However, if anything, the low- versus the high-AGE diet may have led to a less favorable microbiota profile. Most notably, we observed a decreased abundance of *Anaerostipes* after the low- versus the high-AGE diet. Although we hypothesized that a low-AGE diet would improve insulin sensitivity and therefore decrease the risk of future diabetes, a decreased abundance of *Anaerostipes* is suggestive of a more diabetes-prone phenotype. *Anaerostipes* is among the 15 most abundant taxa in healthy individuals [39], and it may use inulin and fructo-oligosaccharides, present in many foods including onions, via trophic chains with the *Bifidobacterium* spp. [40] or even in pure culture (Dr. Nam Bui, unpublished observations) to produce butyrate, a beneficial SCFA that contributes to insulin sensitivity in animal studies [41]. Moreover, a recently discovered propionate-production gene cluster of the *Anaerostipes* species has been associated with beneficial metabolic biomarkers in (pre)diabetes cohorts [42]. One potential explanation for the decreased abundance of the *Anaerostipes* spp. in the low-AGE group may relate to the level of fiber intake, which was slightly albeit statistically significantly higher in the high-AGE group compared to the low-AGE group. Moreover, it is unknown whether the nature of fiber intake was different between these groups as these data were not available. Other potential unfavorable shifts in genera abundance were those of *Tyzerella* and *Family-XII_UCG-001*, which both increased in abundance after the low- versus the high-AGE diet. These genera were enriched in patients with general anxiety disorder [43] and irritable bowel syndrome [44], and ulcerative colitis [45], respectively. In contrast, *Oscillibacter*, which decreased in abundance after the low- versus the high-AGE diet, was enriched in patients with chronic kidney disease and showed positive correlations with uremic metabolites [46]. Of note, a recently discovered *Oscillibacter*-related species, *Dysosmobacter welbionis*, was found to improve insulin sensitivity in mice [47]. *Negativibacillus* has only recently been isolated in humans [48] and has, to our knowledge, not been associated with disease states. Although we observed an increased abundance of the *Christensenellaceae_R-7 Group* after the low- versus the high-AGE diet, and genera of this family are inversely associated with adiposity in humans [49], this comparison only reached statistical significance after exclusion of a non-compliant participant, and its relevance should therefore be interpreted with caution. All in all, it is possible that any beneficial effects of the low-AGE diet on insulin sensitivity were counteracted by less favorable effects on the gut microbiota, ultimately leading to no change in insulin sensitivity after the low- versus the high-AGE diet.

Research on the biological effects of dietary dicarbonyls in humans is scarce, but studies so far are suggestive of beneficial effects. Recently, Maasen et al. showed an inverse association between the greater habitual intake of MGO, but not 3-DG or GO, and a sum score of low-grade inflammation plasma biomarkers in the population-based cohort of the Maastricht Study [manuscript submitted]. How dietary dicarbonyls could exert biological effects is incompletely understood, but options include direct uptake into the circulation [50], endogenous formation of new AGEs, or an effect on the gut microbiota. Regarding the latter, we showed no association between habitual dietary dicarbonyl intake and alpha or beta diversity, but there was a trend of decreased gut microbial richness with the higher intake of 3-DG. This is in agreement with a recent fluorescence in situ hybridization (FISH) analysis showing that of all three dicarbonyls, the antimicrobial capacity of 3-DG was highest [25]. In contrast, a greater habitual intake of 3-DG was associated with a higher abundance of two beneficial genera. Specifically, after adjusting for multiple testing, a higher habitual intake of 3-DG was associated with a higher abundance of *Anaerostipes* and *Fusicatenibacter* (median relative abundance% [IQR] of 1.3% [0.8,2.6]). This genus’ only known species, *Fusicatenibacter saccharivorans*, was reported to be positively associated with production of the anti-inflammatory cytokine IL-10 in ulcerative colitis patients [51]. These combined findings suggest that the beneficial associations between the habitual intake of dicarbonyls may at least partly be mediated by an effect on the gut microbiota composition. Furthermore, as the high-AGE diet in the present study was also higher in dicarbonyls, albeit to a much lesser extent, we cannot exclude the possibility that the increased intake of dicarbonyls in the high-AGE group influenced our results. Although we observed very limited effects of the intervention diets on our outcomes overall, there was an increase in plasma adiponectin after the high- compared to the low-AGE diet [8]. Potentially, this was a result of the increased intake of dicarbonyls rather than AGEs. Interestingly, most associations between the habitual intake of dicarbonyls and genera abundance were not shared between the individual dicarbonyls, suggesting unique relationships. However, all in all, these analyses should be interpreted with caution and mainly serve as a stepping-stone for further research. Although we did adjust for several important potential confounders—age, sex, energy intake, and the Dutch Healthy Diet index—our limited sample size of 72 participants restricted adjustments for additional variables. We also did not measure short and branched chain fatty acids, which could have provided more insight into the biological relevance of these associations as well as the changes in genera abundance after the low- and high-AGE diets. Another limitation, especially regarding the estimations of habitual dicarbonyl intake, is that FFQs may be prone to recall bias [52], and no FFQ so far has been validated for estimating dicarbonyl intake. A final limitation is that the low- and high-AGE diets were not matched for their glycemic load and index. Although doing so is difficult, these factors may influence the endogenous formation of AGEs [53].

To conclude, we report limited consequences of a 4-week diet low or high in AGEs on the gut microbiota composition of abdominally obese but otherwise healthy individuals. The habitual intake of dietary dicarbonyls, especially 3-DG, showed positive and inverse associations with the abundance of several genera. The effects of dietary dicarbonyls on the gut microbiota composition should be evaluated in larger observational studies and randomized controlled trials using a well-controlled dietary intervention.

## 4. Materials and Methods

### 4.1. Study Approval

This study was approved by the Maastricht University Medical Center ethics committee, performed in accordance with the Declaration of Helsinki, and registered at both international and national trial registries (clinicaltrials.gov: NCT03866343, trialregister.nl: NTR7594). All participants provided written informed consent.

### 4.2. Study Population and Design

A total of 82 abdominally obese but otherwise healthy individuals were recruited, as described in detail elsewhere [8]. These participants were randomly assigned to a 4-week dietary intervention low or high in AGEs at a 1:1 ratio in a double blind, parallel design. Randomization was stratified for age (below and above 50 years of age) and sex, in block sizes of 4, and performed by an independent investigator. Both the investigators and participants were blinded to the treatment allocation, and participants were instructed not to inform the investigators about the food items in their dietary intervention. Only the study dietician was aware of the treatment allocation.

### 4.3. Sample Size Calculation

This study was powered to detect a difference in the primary outcome of insulin sensitivity, as described elsewhere [8]. Based on this, 36 individuals per group were needed to detect a statistical difference. Considering a drop-out rate of 12%, we included 41 participants per group, resulting in a total of 82 participants.

### 4.4. Run-In Diet

Prior to the baseline measurement, all participants followed an isocaloric two-day run-in diet. A participants’ habitual energy intake was assessed by a three-day food diary. The run-in diet contained an average amount of dietary AGEs, based on intake in a large population-based cohort [54], and was designed to exclude any influences of high-AGE products consumed in the days prior to the baseline measurement.

### 4.5. Dietary Intervention

Intervention diets were constructed by a trained dietician and were energy- and macronutrient-matched. Both intervention diets adhered to the Dutch dietary guidelines for macro- and micronutrient intake [55] and contained 15 Energy % in calories (En%) protein, 35 En% fat, 48 En% carbohydrates, and 2 En% fibers. With the use of our gold-standard UPLC-MS/MS dietary AGE database that contains approximately 250 food items [1], a theoretical difference of approximately 75% in AGEs was attained between diets while not solely relying on different food preparation methods. Participants prepared their food at home using predefined recurring weekly menus with extensive instructions. Most food items were provided to the participants free of charge by means of a delivery service. Participants were instructed not to change their habitual portion sizes, number of in-between snacks, and the habitual timing of their food intake, not to attempt changes in body weight, and not to consume food supplements during the duration of the study.

### 4.6. Dietary Intake

Adherence to the dietary intervention was measured in three ways. First, participants kept a five-day dietary record in the first and last week of the intervention. Second, participants were additionally contacted in the second and third week of the intervention to assess food intake in a standardized way by a 24-h dietary recall [56]. Nutrient intake from these dietary records and recalls were determined using a nutrient software program (Compl-eat, Human Nutrition Wageningen University, Wageningen, The Netherlands). Third, free AGEs in plasma and 24-h urine samples were compared between groups after the intervention, as described elsewhere [8].

Habitual dietary AGE and dicarbonyl intake was determined by coupling a validated semi-quantative FFQ food frequency questionnaire [57], with a reference period of one year, to our dietary AGE [8] and dicarbonyls [29] database, as described elsewhere [58].

### 4.7. Collection of Stool Samples

Participants were instructed to deposit their stool in a disposable Fecotainer ^®^ (AT Medical B.V., Amsterdam, The Nederlands) within 24-h before their appointment to the lab. These samples were then manually homogenized by a researcher, aliquoted, and frozen at −80 °C until further analysis.

### 4.8. DNA Isolation

DNA was extracted from 200 mg of frozen aliquots of homogenized feces by Repeated Bead Beating (RBB) combined with column-based purification according to the recommended protocol Q of the International Human Microbiome Standards Consortium [59]. Briefly, bead beating was performed using the FastPrep™ Instrument (MP Biomedicals, Santa Ana (CA), USA) with 0.1 mm zirconium-silica beads (BioSpec Products, Bartlesville (OK), USA) to homogenize feces. The DNA was finally purified by adapting it to QIAamp DNA Stool Mini kit columns (Qiagen, Hilden, Germany).

### 4.9. Microbiota Profiling

The V4 region of the 16S rRNA gene was PCR amplified from each DNA sample using the 515F/806R primer pair described previously [60]. After 25 cycles of PCR amplification, amplicons were purified using AMPure XP purification (Agencourt, Beckmann-Coulter, CA, USA) according to the manufacturer’s instructions and eluted in 20 μL 1 × low TE (10 mM Tris-HCl, 0.1 mM EDTA, pH 8.0). Quantification of amplicons was subsequently performed by the Quant-iT PicoGreen dsDNA reagent kit (Invitrogen, Thermofisher, MA, USA) using a Victor3 Multilabel Counter (Perkin Elmer, Waltham, MA, USA). Amplicons were mixed in equimolar concentrations to ensure equal representation of each sample and sequenced on an Illumina MiSeq instrument (Illumina, Eindhoven, The Netherlands) (MiSeq Reagent Kit v3, 2 × 250 cycles, 10% PhiX) to generate paired-end reads of 250 bases in length in both directions.

The pre-processing of sequencing data, using an in-house pipeline based upon DADA2 (R version 4.1.0, R-Project) [61], consisted of the following steps: reads filtering, identification of sequencing errors, dereplication, inference, and removal of chimeric sequences. The length of the raw reads was detected using HTSeqGenie [62], and sequence manipulation was performed using Biostrings [63]. In order to assign taxonomy, DADA2 was used to annotate down to the species level using the database SILVA 138 version 2 [64]. Data were expressed as Amplicon Sequence Variants (ASVs). Decontam was used with the either setting, which combines the two statistical methods’ prevalence and frequency for the identification of contaminating ASVs [65]. Contaminating ASVs identified by decontam were filtered out together with ASVs presented in less than 5% of all samples and ASVs with a total abundance below 0.01% across all samples. Finally, we omitted samples with a low sequencing depth (<40,000 sequences). The final file was saved in the phyloseq format from which the taxa tables were extracted [66].

### 4.10. Gut Microbiota Composition

All gut microbial analyses were performed using the R package “microViz” [67], and species with a prevalence <5% were excluded.

Alpha diversity, a measure of within-sample bacterial diversity, was expressed as both total diversity (richness) and the Shannon index. Total diversity was defined as the total number of individual ASVs within a sample. The Shannon index, a measure of species diversity, takes into account both species abundance and evenness.

Beta diversity, a measure of differences in between-sample bacterial composition, was expressed as both the Aitchison distance and the Bray–Curtis dissimilarity. These beta diversity measures were plotted using principal coordinate analysis (PCoA, for Bray–Curtis dissimilarity) and principal component analysis (PCA, for Aitchison distance) to visualize differences in the bacterial composition between groups before and after the dietary intervention. The Aitchison distance was specifically used to visualize the taxa driving differences in the microbial community structure between groups. Additionally, Bray–Curtis beta diversity was also determined as the within-subject temporal change in the microbial composition due to the low- or high-AGE diet. For beta diversity measures, taxa were aggregated at the genera level.

### 4.11. Statistics

Analyses regarding participant characteristics and dietary intake were conducted using SPSS version 25 for Windows (IBM Corporation, Armonk, NY, USA). Analyses regarding the gut microbiota were conducted using the R package “microViz” [67]. Participant characteristics and dietary intake are presented as means ± SD, medians [interquartile range], or percentages, as appropriate. Differences in dietary intake between both groups were assessed by a one-factor ANCOVA while adjusting for sex, age, and energy intake. Differences in alpha diversity between both groups were assessed by a one-factor ANCOVA while adjusting for sex, age, and alpha diversity at baseline. Differences in overall gut microbial composition (i.e., centroid of both groups’ beta diversity in the PCoA and PCA plots) were statistically tested with permutational multivariate analyses of variance (PERMANOVA). Differences in the specific gut microbial composition after the intervention (i.e., differential abundance analysis) were assessed using beta binomial regression aggregated at the genus level, adjusted for age and sex. This was also performed at baseline to rule out differences already present before the intervention. To account for multiple testing, we applied the Benjamini–Hochberg (B–H) multiple comparison correction with a false discovery rate of 10%. Results of beta binomial regression, being fold-changes between intervention diets, were visualized in a taxonomic association three. Subsequently, genera that were differentially abundant after the low- versus the high-AGE diet were transformed to relative abundance and visualized as box plots. To investigate associations between the habitual intake of dicarbonyls and measures of alpha diversity, we used multiple linear regression, adjusting for potential confounders in two separate models. In model 1, we adjusted for age and sex. In model 2, we adjusted for energy intake and the Dutch Healthy Diet score [68], in addition to age and sex. To investigate associations between the habitual intake of dicarbonyls and taxonomic abundance, we used beta binomial regression while adjusting for age, sex, energy intake, and the Dutch Healthy Diet Index. A *p*-value p of <0.05 was considered statistically significant.

## Figures and Tables

**Figure 1 ijms-23-05328-f001:**
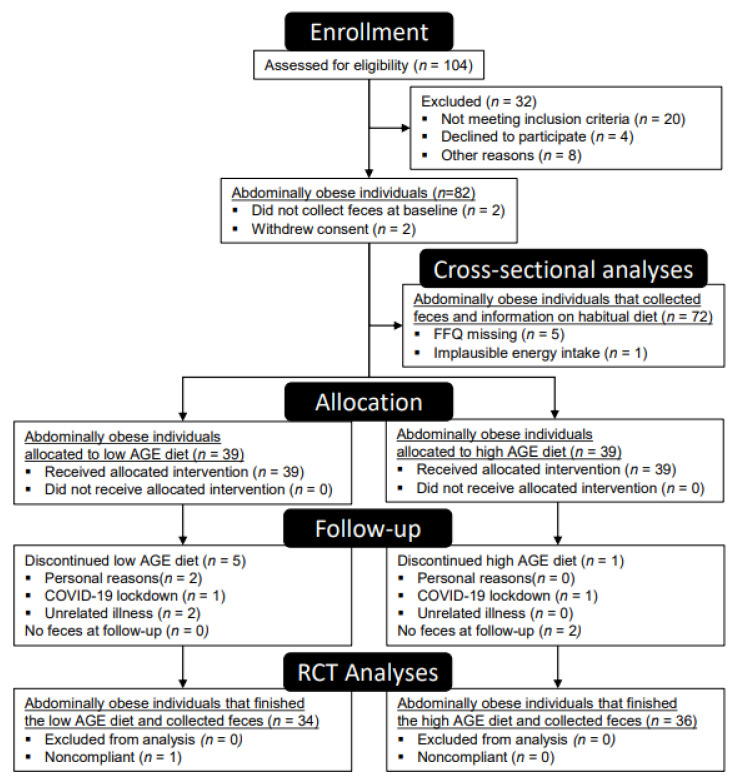
CONSORT flow diagram for RCT and cross-sectional analyses.

**Figure 2 ijms-23-05328-f002:**
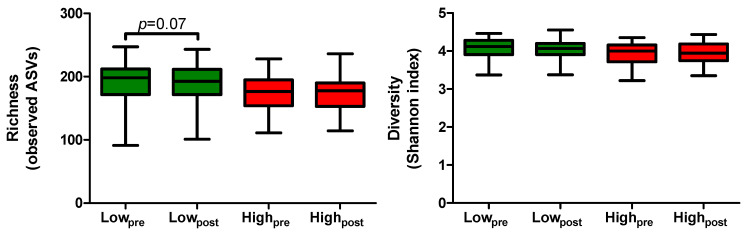
Richness (**left**) and gut microbial diversity (**right**) before and after a 4-week diet low or high in AGEs. Sample sizes: low-AGE group *n* = 34, high-AGE group *n* = 36. Within-group differences were tested with a paired samples *t*-test. Treatment effects were tested with a one-way ANCOVA with adjustment for age, sex, and the baseline variable of interest.

**Figure 3 ijms-23-05328-f003:**
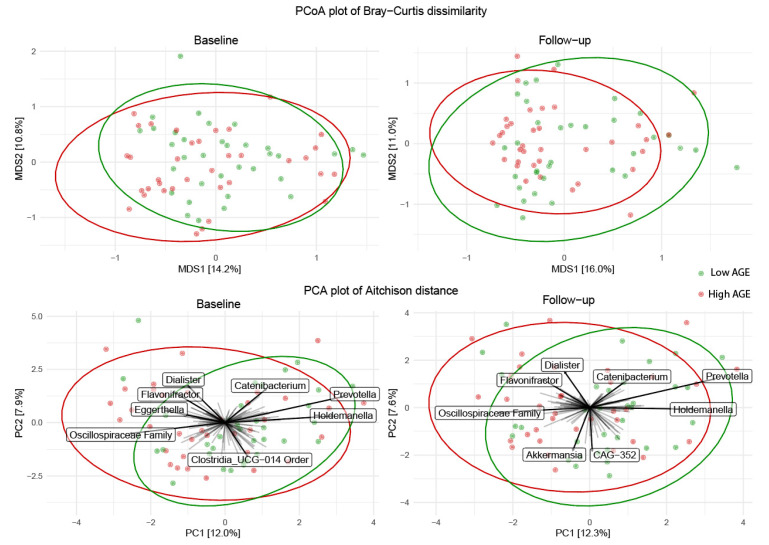
Measures of beta-diversity before (**left** column) and after (**right** column) a low- or high-AGE diet in abdominally obese individuals. Upper row: principle coordinate analysis of Bray–Curtis dissimilarity. Lower row: Principle component analysis of the Aitchison distance. Sample sizes: low-AGE group *n* = 34, high-AGE group *n* = 36.

**Figure 4 ijms-23-05328-f004:**
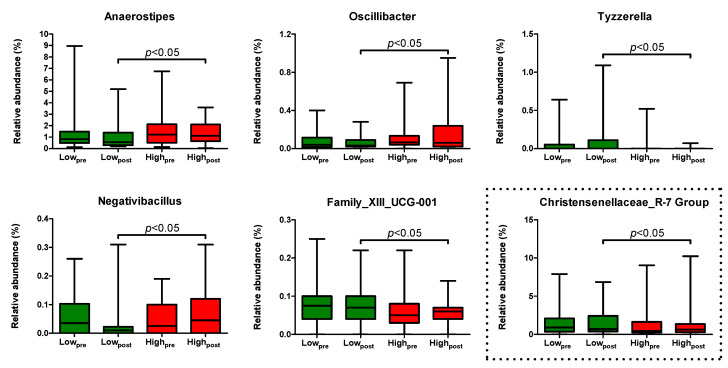
Relative abundance of differentially abundant genera after a 4-week low- (in green) or high-AGE diet (in red) in abdominally obese individuals. *n* = 34 for the low-AGE group and *n* = 36 for the high-AGE group. Statistical significance was assessed using beta binomial regression with adjustments for age and sex. Please note that the difference in the relative abundance of the *Christensenellaeceae_R-7 Group* only became statistically significant after exclusion of a non-compliant participant of the low-AGE group. This participant was included in all other comparisons. All comparisons became statistically non-significant after correction for multiple testing.

**Table 1 ijms-23-05328-t001:** Characteristics of 70 abdominally obese individuals included in the deAGEing trial at baseline.

Characteristic	Low AGE (*n* = 34)	High AGE (*n* = 36)
**Demographics**		
Age (years)	52 ± 13	54 ± 13
Males/Females	10/24	11/25
Weight (kg)	87.7 ± 14.3	88.0 ± 13.1
Waist circumference (cm)		
Men	106.7 ± 4.8	107.5 ± 7.1
Women	101.2 ± 8.6	100.1 ± 8.2
BMI (kg·m^−2^)	30.4 ± 4.1	30.8 ± 4.2
24-h systolic BP (mmHg) ^1^	126 ± 13	124 ± 9
24-h diastolic BP (mmHg) ^1^	80 ± 9	77 ± 7
**Biological**		
Fasting glucose (mmol/L)	4.9 ± 0.4	5.1 ± 0.5
Total cholesterol (mmol/L)	5.0 ± 0.9	5.4 ± 0.8
LDL cholesterol (mmol/L)	3.3 ± 0.9	3.7 ± 0.7
HDL cholesterol (mmol/L)	1.4 ± 0.4	1.3 ± 0.3
Triglycerides (mmol/L)	1.2 ± 0.4	1.6 ± 0.7
**Feces**		
Richness (observed species)	194 ± 30	173 ± 32
Shannon index	4.08 ± 0.27	3.93 ± 0.30
Bristol stool scale	4 ± 1	4 ± 1

Data are presented as means ± SD. ^1^ Low-AGE *n* = 32, High-AGE *n* = 35.

**Table 2 ijms-23-05328-t002:** Average daily AGE, dicarbonyl, energy, and macronutrient intake of 70 abdominally obese individuals during the low- or high-AGE dietary intervention.

Nutrient	Low AGE (*n* = 32) ^1^	High AGE (*n* = 36)	Low vs. High *p*
**AGEs (mg/day)**			
CML	2.68 ± 0.67	6.90 ± 1.32	<0.001
CEL	1.72 ± 0.40	8.94 ± 1.98	<0.001
MG-H1	13.67 ± 3.11	48.75 ± 11.93	<0.001
**Dicarbonyls (mg/day)**			
MGO	3.04 ± 0.89	3.76 ± 1.00	<0.001
GO	2.84 ± 0.73	3.20 ± 0.70	<0.001
3-DG	13.86 ± 5.33	19.15 ± 5.88	<0.001
**Energy (kcal/day)**			
Energy intake ^2^	2034 ± 476	2078 ± 471	0.612
**Macronutrients (energy %)**			
Protein	17.1 ± 1.6	16.7 ± 1.5	0.325
Plant-based protein	6.4 ± 0.8	7.6 ± 0.6	<0.001
Animal-based protein	10.7 ± 1.8	9.1 ± 1.6	<0.001
Fat	31.6 ± 2.6	35.6 ± 3.0	<0.001
Saturated fat	12.8 ± 1.5	12.0 ± 0.8	0.009
Mono-unsaturated fat	9.7 ± 0.8	12.7 ± 1.6	<0.001
Poly-unsaturated fat	6.1 ± 1.1	7.7 ± 1.5	<0.001
Carbohydrates	48.4 ± 2.7	44.7 ± 2.8	<0.001
Mono- and disaccharides	21.2 ± 2.8	19.4 ± 2.7	0.008
Polysaccharides	27.2 ± 2.3	25.3 ± 1.5	<0.001
Fiber	2.1 ± 0.2	2.3 ± 0.1	0.001
Alcohol	0.0 [0.0,0.60]	0.0 [0.0,0.76]	0.966

Daily intakes (means ± SD, medians [IQR]) were assessed from two five-day dietary logs in week one and week four of the intervention. Differences between intervention groups were tested by a one-factor ANCOVA with energy intake, sex, and age as covariates, and differences in alcohol intake were tested by the non-parametric Mann–Whitney U test. ^1^ Dietary logs were not returned by two participants in the low-AGE group. ^2^ Energy intake was not included as a covariate.

## Data Availability

Data described in the manuscript are available from the corresponding author upon request pending application and approval.

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
