# Peer review of "A 4-Week Diet Low or High in Advanced Glycation Endproducts Has Limited Impact on Gut Microbial Composition in Abdominally Obese Individuals: The deAGEing Trial"

_ijms, 2022, doi:10.3390/ijms23105328_

Round 1

Reviewer 1 Report

Ln.49 – when citing this publication of Koschinsky, it is necessary to stress that the method of AGE determination differs substantially from that used by the authors. Is there any knowledge on the absorption of individual AGEs?

Ln.80 – Lit #26 is the corrigendum to lit #29. Is there any reason to cite the corrigendum?

Ln.90-92 – please delete.

Figure 2 – I wonder why there is a difference in richness between the groups at the beginning of the intervention. Is this difference significant and if so, why might that be the case?

Ln.221 – a word seems to be missing.

Ln.287 – do genes produce propionate?

Ln.303 – “Christensenellaceae”

Ln.325 – is it not “saccharivorans”?

Author Response

Response to Reviewer 1 Comments

We thank the reviewer for his/her feedback.

Point 1: Ln.49 – when citing this publication of Koschinsky, it is necessary to stress that the method of AGE determination differs substantially from that used by the authors. Is there any knowledge on the absorption of individual AGEs?

Response 1: This is a justified remark by the reviewer and this statement deserves nuance. There are no direct studies in humans on differences in absorption of individual dietary AGEs (in our case CML, CEL, and MG-H1). However, based on its higher abundance in food, levels of free MG-H1 in plasma are also higher than those of CML and CEL following a high-AGE diet. 

Based on the above, Ln 49 has been revised into: “Although the metabolism of (individual) dietary AGEs is still largely unknown, early estimations using ELISA techniques suggest that only 10% of dietary AGEs are absorbed into the circulation, so that the majority should pass through the colon.”

Point 2: Ln.80 – Lit #26 is the corrigendum to lit #29. Is there any reason to cite the corrigendum?

Response 2: There is no reason to cite the corrigendum. Citation 26 has been removed and replaced with citation 29.

Point 3: Ln.90-92 – please delete.

Response 3: Deleted.

Point 4: Figure 2 – I wonder why there is a difference in richness between the groups at the beginning of the intervention. Is this difference significant and if so, why might that be the case?

Response 4: Important determinants of richness (such as age, use of antibiotics, obesity severity) were not different between both groups. Furthermore, by design, participants were healthy (no cardiovascular disease, inflammatory bowel disease and hypertension). As such, we are unable to explain this observation. Ultimately, any differences observed at baseline should occur at random due to the randomization procedure. The consequence of the difference in richness at baseline should be limited as this variable is adjusted for in the statistical models.

         Nonetheless, the difference in richness at baseline is statistically significant (mean difference of 18 ASVs, p = 0.012, tested with a two-way unpaired samples t test). We chose not to report this statistic as testing for differences at baseline in a randomized controlled trial is not supported by the CONSORT statement.   

Point 5: Ln.221 – a word seems to be missing.

Response 5: In fact, the word “and” was unnecessary.

Point 6: Ln.287 – do genes produce propionate?

Response 6: This was a spelling mistake and the sentence has been adapted.

Point 7: Ln.303 – “Christensenellaceae”

Response 7: Adapted.

Point 8: Ln.325 – is it not “saccharivorans”?

Response 8: The reviewer is right and this has been adapted. 

Reviewer 2 Report

The authors present a very well-structured work of microbiota analysis, it is rare to see the application of the Bristol scale! Factor that can greatly alter the results obtained.

I would propose to:

- Have a more complete view of the AGEs and carbonyls present in foods, or how they were determined in the course of the 4 weeks? Is it just an estimate regarding the reports or were there any analyzes? Because if this is the only way it is a limitation of the study that should at least be underlined, I understand that it would be impossible to determine the AGEs step by step, but it is obvious that this implies an approximation that could also be very large, given that the AGEs are very correlated to temperature. and the cooking time of the food

- Would it be possible to have an analysis of the AGEs and carbonyls in the feces? In order to evaluate how many actually reach and influence the microbiota.

- line 379 I guess EN means percentage in calories, I would put it explicitly and fiber does not provide available calories

- finally, I know it is complicated, but glycemic variations should also be standardized since these are a source of endogenous AGEs that are added to exogenous ones

Author Response

Response to Reviewer 2 Comments

The authors present a very well-structured work of microbiota analysis, it is rare to see the application of the Bristol scale! Factor that can greatly alter the results obtained.

We thank the reviewer for his/her compliments and constructive criticism.

I would propose to:

Point 1: Have a more complete view of the AGEs and carbonyls present in foods, or how they were determined in the course of the 4 weeks? Is it just an estimate regarding the reports or were there any analyzes? Because if this is the only way it is a limitation of the study that should at least be underlined, I understand that it would be impossible to determine the AGEs step by step, but it is obvious that this implies an approximation that could also be very large, given that the AGEs are very correlated to temperature and the cooking time of the food

Response 1: The reviewer is correct that the AGE and dicarbonyl content of both the intervention diets and the habitual diet is based on estimates. Although participants received extensive written and oral instructions on how to prepare their meals, and several products from the intervention diets were ready-to-eat, the actual AGE content may differ due to temperature and cooking time. The reviewer is also correct that a step-by-step determination of AGE content of all foods in the intervention diet is impossible due to practical reasons.

         Nonetheless, we believe that for the low- and high-AGE diet this is of less importance because the large difference in AGE content between diets is confirmed by dietary logs, 24-hour recalls, and plasma and urinary concentrations of all three free AGEs. This information is presented in our previously published manuscript regarding the main outcomes of this trial [1]. For the dicarbonyl content of the habitual diet, assessed by a food frequency questionnaire, this limitation is of greater relevance. As such, we have now underlined this limitation in the discussion section: “Another limitation, especially regarding the estimations of habitual dicarbonyl intake, is that FFQs may be prone to recall bias [2] and no FFQ so far has been validated for estimating dicarbonyl intake.”

Point 2: Would it be possible to have an analysis of the AGEs and carbonyls in the feces? In order to evaluate how many actually reach and influence the microbiota.

Response 2: While this analysis is technically possible, we believe it will serve limited purpose for this specific question. It has already been shown that CML content of feces is increased after a high-AGE diet (Delgado-Andrade et al [3], secondary analyses of ref 20 from the original manuscript). As such, we would expect a similar pattern in our analyses, especially given the confirmed increase in free AGEs in plasma and urine after the high AGE diet.

More importantly, increased levels of CML, CEL, and MG-H1 in feces may not necessarily depict a greater interaction with the gut microbiota. As mentioned in the introduction, it is hypothesized that dietary AGEs may act as substrates for gut microbes. The measurement of the unknown metabolites formed in this process may provide a better reflection of the interaction with the gut microbiota, but more research is needed for their identification.

Point 3: Line 379 I guess EN means percentage in calories, I would put it explicitly and fiber does not provide available calories

Response 3: Indeed, En% means energy percentage in calories. This has now been written in full to avoid confusion.

We attributed 2kcal per gram of fiber based on reports of the Food and Agricultural working group [4]. Although the net energy contribution of fibers may vary by individual (being dependent on fermentation, satiety, among others), this estimate has also been adapted by the European Union and is used in the Dutch National food composition table, on which our calculations are based. Ultimately, our intention is to show a directly comparable estimate of fiber intake between groups, regardless of total energy intake. 

Point 4: Finally, I know it is complicated, but glycemic variations should also be standardized since these are a source of endogenous AGEs that are added to exogenous ones.

Response 4: The reviewer is right in that glycemic variations may add to the endogenous formation of AGEs and should therefore by accounted for as much as possible. Our intervention diet was specially designed based on our dietary AGE database while matching total energy intake and macronutrient intake as much as possible. As the available food products in our dietary AGE database are limited (around 250 products), additionally controlling for glycemic index and load was not possible. We recognize this limitation and an explanation has been added to the discussion section.

References

  1. Linkens, A.M.A.; Houben, A.J.; Niessen, P.M.; Wijckmans, N.; de Goei, E.; Van den Eynde, M.D.; Scheijen, J.; Waarenburg, M.; Mari, A.; Berendschot, T.T.; et al. A 4-week high-AGE diet does not impair glucose metabolism and vascular function in obese individuals. JCI Insight 2022, doi:10.1172/jci.insight.156950.
  2. Naska, A.; Lagiou, A.; Lagiou, P. Dietary assessment methods in epidemiological research: current state of the art and future prospects. F1000Res 2017, 6, 926, doi:10.12688/f1000research.10703.1.
  3. Delgado-Andrade, C.; Tessier, F.J.; Niquet-Leridon, C.; Seiquer, I.; Pilar Navarro, M. Study of the urinary and faecal excretion of Nε-carboxymethyllysine in young human volunteers. Amino acids 2012, 43, 595-602, doi:10.1007/s00726-011-1107-8.
  4. Maclean, W.; Harnly, J.; Chen, J.; Chevassus-Agnes, S.; Gilani, G.; Livesey, G.; Warwick, P. Food energy–Methods of analysis and conversion factors. In Proceedings of the Food and agriculture organization of the united nations technical workshop report, 2003; pp. 8-9.

Round 2

Reviewer 2 Report

I think that that authors made enough improvement to make the manuscript published